# Combining Flow Cytometry and Metagenomics Improves Recovery of Metagenome-Assembled Genomes in a Cell Culture from Activated Sludge

**DOI:** 10.3390/microorganisms11010175

**Published:** 2023-01-10

**Authors:** Nafi’u Abdulkadir, Joao Pedro Saraiva, Florian Schattenberg, Rodolfo Brizola Toscan, Felipe Borim Correa, Hauke Harms, Susann Müller, Ulisses Nunes da Rocha

**Affiliations:** 1Department of Environmental Microbiology, Helmholtz Center for Environmental Research, Permoserstrasse 15, 04318 Leipzig, Germany; 2Department of Biochemistry, Faculty of Natural Science, University of Leipzig, Bruderstrasse 34, 04103 Leipzig, Germany

**Keywords:** cell sorting, flow cytometry, metagenome-assembled genomes, metagenomics

## Abstract

The recovery of metagenome-assembled genomes is biased towards the most abundant species in a given community. To improve the identification of species, even if only dominant species are recovered, we investigated the integration of flow cytometry cell sorting with bioinformatics tools to recover metagenome-assembled genomes. We used a cell culture of a wastewater microbial community as our model system. Cells were separated based on fluorescence signals via flow cytometry cell sorting into sub-communities: dominant gates, low abundant gates, and outer gates into subsets of the original community. Metagenome sequencing was performed for all groups. The unsorted community was used as control. We recovered a total of 24 metagenome-assembled genomes (MAGs) representing 11 species-level genome operational taxonomic units (gOTUs). In addition, 57 ribosomal operational taxonomic units (rOTUs) affiliated with 29 taxa at species level were reconstructed from metagenomic libraries. Our approach suggests a two-fold increase in the resolution when comparing sorted and unsorted communities. Our results also indicate that species abundance is one determinant of genome recovery from metagenomes as we can recover taxa in the sorted libraries that are not present in the unsorted community. In conclusion, a combination of cell sorting and metagenomics allows the recovery of MAGs undetected without cell sorting.

## 1. Introduction

Metagenomics is a standard technique used to study microbial community composition and functional potential [1,2,3,4,5]. Thanks to novel bioinformatics tools, reconstructing genomes in complex communities has significantly improved [1]. High-throughput sequencing allows analyzing genomes from species within microbial communities, including hitherto-uncultured species [6,7,8]. The potential for obtaining genomes of uncultivable species with metagenomics has been proven by assembling nearly complete MAGs of dominant species from microbial communities with relatively low microbial diversity [9]. Nevertheless, the recovery of metagenome-assembled genomes (MAGs) is currently challenged by a lack of understanding of how specialized bioinformatics tools for assembly and binning influence MAG recovery [10]. For example, insufficient sequencing depth makes it difficult to reconstruct MAGs of individual species with relative abundances lower than 1% in a given metagenome [9]. This makes the recovery of MAGs of low-abundance species in complex communities difficult, as the reconstruction of nearly complete MAGs may demand hundreds of millions of short-sequencing reads (150–200 bp) per metagenome [11,12].

The limitations of current metagenomics approaches, such as access to rare species, introduce bias to the taxonomic composition of complex microbial communities in genome-centric metagenome studies. Eliminating this bias is particularly interesting in wastewater treatment plants because rare species, such as nitrifiers, may be underestimated or ignored. The efficiency of the system depends on the presence of certain species in the activated sludge (AS) [13], which is the most used process for wastewater treatment globally [13] and consists of a high concentration of microorganisms able to convert organic and inorganic wastewater constituents, thereby purifying water and synthesizing resources [13] that can be used in biotechnological processes [14]. Additionally, previous studies suggest that low abundant taxa found in a complex environment may become relevant back-ups under environmental stress [15] and should not be overlooked.

A potential strategy to circumvent the challenges of recovering low-abundant species in metagenomes is to reduce the complexity of a target microbial community. Greib and collaborators [12] revealed that obstacles associated with access to low abundant taxa in complex communities using metagenomics and single-cell sequencing could be addressed. To this end, microbial target groups were enriched before genomic analysis by taxon-specific fluorescent labeling of microbial species and fluorescence-activated cell sorting (FACS). FACS seems to be a promising technology that separates microbial communities into sub-communities [16] that can be sequenced separately. Flow cytometry and cell sorting have been combined as a high throughput technique for analyzing microbiomes at the single-cell level [17,18]. Cell sorting does not impair cells when separating them from the whole community [19], and reduces contamination by extracellular DNA [20]. The usefulness of this approach has already been demonstrated in soil samples [20]. The study of Alteio et al. [20] demonstrated that cell sorting and metagenomics approaches expanded the diversity of soil taxa and recovered more genomes from ‘minimetagenomes’ than bulk metagenomes.

This study aimed to improve MAG recovery by combining flow cytometry, cell sorting, and metagenomics from activated sludge microbiomes. Briefly, we sequenced metagenomes of the unsorted and sorted communities from a batch-grown activated sludge microbiome collected in a wastewater treatment plant. After MAG recovery, we dereplicated the different MAGs into genome operational taxonomic units (gOTUs) and compared their phylogeny and abundances to 16S rRNA genes (rOTUs) reconstructed from the different libraries. We used our data to check if the recovered MAGs belonged to the dominant microbial community in our cell culture. In addition to improving genome recovery from sorted and unsorted communities, we used our dataset to extrapolate the factors controlling the recovery of MAGs, including genome coverage and relative abundances. Furthermore, we defined a detection limit for MAG recovery and a strategy to define cut-off limits for MAG recovery in metagenomes.

## 2. Materials and Methods

### 2.1. Cell Cultivation

We used the cells of a microbial community originating from an activated sludge basin of a wastewater treatment plant (Eilenburg, Saxonia, Germany—51°27’39.4″ N, 12°36’17.5″ E) as our model microbial community described in Liu et al. [21]. The entire workflow employed in this study, including the experimental scheme of the sequential batch cultivation and treatment of the microbial community, is presented in Appendix A. The activated sludge sample (30 mL thawed inoculum) was first cultured in a 1-L batch flask in 300 mL medium at 30 °C and 125 rpm for 17 h to reduce the undegraded and inorganic particles from the sludge. The second and third batches (each also in 1-L flasks) were inoculated with an initial optical density (OD) of 0.05 (OD_600nm_, d = 0.5 cm), each, and cultivated in the same way.

The medium was a mixture of 98% synthetic wastewater and 2% peptone. The medium constituted 0.198 g L^−1^ peptone (from meat), 0.2 g L^−1^ meat extract, 0.219 g L^−1^ yeast extract, 0.1 g L^−1^ glucose, 0.49 g L^−1^ Na-propionate (filtered), 0.0059 g L^−1^ CaCl_2_·2H_2_O, 0.0294 g L^−1^ KCl, 0.06 g L^−1^ NaCl, 0.04 g L^−1^ K_2_HPO_4_, 0.2156 g L^−1^ KH_2_PO_4_ and 0.0196 g L^−1^ MgSO_4_·7H_2_O, purchased from: Merck KGaA (Darmstadt, Germany), SERVA Electrophoresis GmbH (Heidelberg, Germany) and Carl Roth GmbH (Karlsruhe, Germany). 

### 2.2. Cell Fixation 

The wastewater microbial community samples were fixed and stained for cytometric analyses, as described by Liu et al. [21]. Briefly, 4 mL of sample from each of the three batches (see Section 2.1) was taken and centrifuged (3200× *g* at 4 °C) for 10 min. The supernatant was discarded. We washed the cells with phosphate-buffered saline once (PBS, 6 mM Na_2_HPO_4_, 1.8 mM NaH_2_PO_4_, 145 mM NaCl, pH 7, 3200× *g*, 10 min, 4 °C) and stabilized them by adding 2 mL paraformaldehyde solution (PFA, 2% in PBS) to the cell pellet and incubated for 30 min at room temperature (RT). Next, the samples were centrifuged (3200× *g*, 10 min, 4 °C), re-suspended in 4 mL of EtOH (70%), and then stored at −20 °C. 

### 2.3. Cell Staining 

We washed the stored samples twice by centrifugation (3200× *g*, 10 min, 4 °C) with PBS. The cell solutions were adjusted to an OD of 0.035 (d_ʎ700nm_ = 0.5 cm) using PBS. A quantity of 2 mL of an adjusted sample solution was centrifuged (3200× *g*, 10 min, 4 °C). The pellet was re-suspended with 1 mL of solution A (0.11 M citric acid and 4.1 mM Tween 20, with distilled water) and incubated at RT for 10 min in an ultrasonication bath (35 kHz, Merck Eurolab, Darmstadt, Germany) and 10 min without any further treatment. After another centrifugation step (3200× *g*, 10 min, 4 °C), the cells were stained with 2 mL solution B [0.24 μM DAPI (4’,6-diamidino-2-phenylindole) in phosphate buffer (289 mM Na_2_HPO_4_ and 128 mM NaH_2_PO_4_ in distilled water)] overnight at RT in the dark.

### 2.4. Flow Cytometric Measurement 

Samples were measured with the BD Influx v7 Sorter (Becton, Dickinson and Company, Franklin Lakes, NJ, USA) to visualize the community structure as described by Liu et al. [21]. Samples from the third batch were used for cell sorting (Appendix A). Briefly, the cell data was collected in a 2D cytometric histogram according to DAPI (4′, 6′-diamidino-2-phenylindole) fluorescence and forward scatter (FSC). DAPI gives information on DNA contents, and FSC provides information related to cell size. A cell gate was created, which comprised 200,000 cells per measurement. Cell sorting gates were generated by labeling clusters of cells in the 2D plots that differed based on the respective optical properties of the cells [17]. We generated a total of 21 gates from the cell gate template. The 21 sorted gates had an average cell abundance per gate (7658.1) and were distributed into two pooled sub-communities (Appendix A). 

### 2.5. Cell Sorting of Sub-Communities

Cell sorting was conducted as described by Koch et al. [17]. Briefly, sub-communities were selected and sorted at a rate of 5000 cells per second. To reach the targeted amount of at least 50 ng DNA per 20 µL for metagenomics analysis, cells were sorted into a plastic tube with a maximum of 2.25 × 10^6^ cells per tube. We tested the number of cells needed and reached the value of 4.95 × 10^7^ cells (>50 ng). Therefore, it was necessary to pool 22 tubes per sub-community. Next, we concentrated these tubes (22 per sub-community) by centrifuging twice the sorted cells. We first centrifuged the cells (20,000× *g*, 25 min, 6 °C) and kept only 80 µL in each tube. The cells were vortexed for 10 s and pipetted up and down several times before pooling them into one tube. Next, we centrifuged (20,000× *g*, 25 min, 6 °C) the cells and discarded the supernatant completely. We stored the pellet directly at −20 °C.

Gates were affiliated based on their cell number in relation to the average cell number per gate (7658.1 cells). Dominant (DG) sub-communities consist of gates whose cell counts are above the average cell number per gate, while low abundant (LA) sub-communities consist of gates whose cell numbers are below the average cell number per gate. Additionally, Outer (OG) sub-communities are those in the master cell gate template, but outside of DG and LA sub-communities (OG = master cell gate − (DG + LA)), Appendix A). The unsorted microbial community (hereafter, UC) was used as a control.

### 2.6. DNA Extraction and Sequencing

We extracted DNA from the UC and the three sub-communities (DG, LA, and OG) using the Chelex method as described by Koch et al. [17]. In short, the unsorted and sorted pelleted cells were stored at −20 °C before the extraction. Next, we added 70 µL of 10 % Chelex (Biorad, Hercules, CA, USA) solution under a sterile bench to the pellet of sorted sub-communities and 300 µL to the pellet of the unsorted community. The cells were vortexed for 5 s and kept in 2 mL tubes for 45 min at 95 °C using a thermocycler (Biorad, Hercules, CA, USA). Next, samples were centrifuged for 5 min at 7000× *g* and 4 °C. Finally, the supernatant (50 µL) was carefully collected into a sterile DNA-free tube without any Chelex beads. The DNA concentration was measured with Qubit^®^ 3.0 (Life Technologies, Carlsbad, CA, USA) after DNA extraction. The pellet was stored at −20 °C before library preparation and sequencing. The extracted genomic DNA was submitted to StarSEQ^®^ GmbH (Mainz, Germany) for Illumina library preparation and sequencing. 

### 2.7. Metagenome Libraries, Quality Control, and Assembly of Raw Reads

The DNA extracts from the four metagenomes (UC, DG, LA, and OG) were used for library preparation and sequencing. The libraries were prepared using the Nextera XT DNA preparation kit from Illumina according to the manufacturer’s instructions. Next, metagenomes were paired-end sequenced using Illumina (Illumina NextSeq 500; 2 × 150 bp) at a minimum of 20 million reads per library. Finally, high-throughput sequencing was done by StarSEQ^®^ GmbH. The sequenced libraries were imported in FASTQC format. The reads were quality controlled by removing low-quality reads and adapters using Trim Galore v0.4.3 implemented in metaWRAP pipeline [22]. The trimmed reads were assembled to contigs using metaSPAdes v3.11.1 [23]. 

### 2.8. Recovery of Metagenome-Assembled Genomes (MAGs) 

Prokaryotic MAGs were generated using the metaWRAP v0.7 pipeline [22] with metaBAT v2.12.1 [24], Maxbin v2.2.4 [25], and CONCOCT v0.4.0 [1] as binning tools. The refinement of MAGs was performed using the metaWrap binning_refiner v1.2 [22]. MAG quality was determined using CheckM v1.1.6 [26], and the quality score was calculated based on completeness and contamination values (Equation (1)) as defined by Park et al. [10].
(1)$$quality\;score=completeness\;\left(\%\right)-\left(5\times contamination\;\left(\%\right)\right)$$

In Equation (1), completeness is the estimation of genome quality based on the presence or absence of single-copy marker genes, and contamination is the evaluation of genome quality as revealed by multiple copies of marker genes [26].

MAGs were classified as medium quality if their quality score was greater than 50 and with completeness between 50% and 80% and less than 10% contamination. MAGs with a quality score greater than 50, completeness above 80%, and contamination below 5% were considered high-quality.

### 2.9. Phylogenetic Classification

We used GTDB-tk v0.3.2 [27,28] to assign the taxonomy to the MAGs recovered from all libraries. We selected a representative MAG from each taxonomic group for relative abundance analysis. This selection was based on the average nucleotide identity (ANI) distance between microbial species’ genomes clustered at 95% ANI and on quality score metrics. We used Fastree2 [29] (default parameters) to construct a phylogenetic tree with concatenated protein alignments of gOTUs recovered from the unsorted community and sorted sub-communities. We used iTol [30] to generate the visual representation of the tree via the EMBL server (https://itol.embl.de/, accessed on 8 September 2022).

### 2.10. Clustering of MAG to Genome Operational Taxonomic Units (gOTUs)

To assess a better representation of our genomes across sorted sub-communities and unsorted communities, we dereplicated the recovered MAGs into gOTUs using MuDoGeR [31]. Briefly, MAGs and their reference sequences classified to the same taxonomy were grouped and clustered into gOTUs at 95% average nucleotide identity (ANI) distance using fastANI v1.0 [32], with default parameters. Next, we used hierarchical clustering with bootstrap resampling, resulting in 13 gOTUs with unique taxonomic classifications. Next, the gOTUs with the best quality per taxonomic cluster were selected as representatives and further processed for relative abundance analysis. To note that MAGs classified as *Sphingobacterium* (at the genus level) recovered in the DG and *Comamonas B*-9 (classified at the species level according to EZBioCloud.net) recovered in the UC were not included in further analyses. 

### 2.11. Calculation of Genome Operational Taxonomic Units Relative Abundances and Coverage

The quality-checked reads from the four libraries were mapped to the gOTUs even if they were not recovered in a given library. Read mapping for all libraries was done using Bowtie2 [33], and mapped reads were retrieved using Samtools [34]. The relative abundance of each gOTU was calculated using Samtools [34]. We calculated the coverage values of gOTUs by multiplying the number of reads of genomes by the average size of reads in the libraries divided by the size of genomes (in base pairs) (Equation (2)). This analysis allowed a comparison of gOTUs abundances across different libraries. The evolutionary history of prokaryotes is complex, and it has been suggested that lateral gene transfer happens more often than not [35], which could lead to the recovery of genomes from species that are not present in the community. To avoid partial detection of (false positive) gOTUs in samples where they were not present, we assigned a gOTU in a sample when it showed a minimum of 10x coverage. The gOTUs with a coverage smaller than 10x were considered below detection limit (BDL). We also performed rarefactions analysis on the metagenomics libraries to determine if we reached the maximum detection of gOTUs within our detection limit by plotting the sample sizes versus observed coverage using the Vegan package of R [36].
(2)Coverage=L∗N/G

In Equation (2), L is the average size of reads per library, *N* is the number of reads per genome and *G* is the genome size (in base pairs).

### 2.12. Reconstruction of Phylogenetic Marker Genes from Metagenomic Libraries

We reconstructed nearly full-length phylogenetic marker genes (16S rRNA) from metagenomic libraries using the Mapping-Assisted Targeted Assembly for Metagenomics (MATAM) pipeline v1.6.0 [37]. The paired-end reads of the metagenomic libraries were interleaved, and taxonomic information of 16S rRNA sequences was obtained by alignment to the SILVA database version 132 NR95 [38] with the RDP classifier [39]. 

Furthermore, MATAM mapped the reads of each marker gene to the metagenomic libraries and calculates the relative abundances of each 16S rRNA reconstructed sequence. 

### 2.13. Curation of Reconstructed 16S rRNA, Removal of Chimeric and Clustering of Sequences

After running MATAM, we removed all 16S rRNA sequences smaller than 900 bp using SeqKit [40] since these might not identify bacteria at the species level. Next, chimeric sequences were removed from 16S rRNA sequences using the reference database SILVA 132 NR95 with the uchime_ref function of UCHIME [41]. Next, we performed primary clustering at 97% similarities per library. Furthermore, we merged representative 16S rRNA sequences from all libraries resulting from the primary clustering and clustered them at 97% similarity using cluster fast function from VSEARCH [42]. We name this final set of reconstructed 16S rRNA genes clustered at 97% similarity to our rOTUs. Additionally, to determine which rOTUs map to the gOTUs, sequences of rOTUs reconstructed from MATAM were further classified to species level using EZbiocloud [43]. 

### 2.14. Calculation of Relative Abundances of rOTUs

The relative abundances of rOTUs were determined in three steps. First, the relative abundances of 16S rRNA sequences obtained from MATAM were calculated by dividing the counts of each 16S rRNA sequence by the total number of sequences in each library (Appendix A). Second, from the primary clustering, we summed up the relative abundances of all sequences in each cluster per library (Appendix A). In the last step, we determined the relative abundances of rOTUs. After, we summed them with all other rOTUs in the clusters resulting from the secondary clustering (secondary clustering was performed with the merged representative sequences from all libraries (Appendix A)).

## 3. Results

### 3.1. Separation of Sub-Communities by Flow Cytometry 

A wastewater microbial community was cultivated in sequential batch culture prior to cytometric analyses. In a cytometric histogram, a total of 200,000 cells per sample were measured using cell information on scattering light and fluorescence signals attached to the DNA contents of cells (Figure 1). A total of twenty-one gates were generated inside of the cell gate according to the position and cell abundances, reflecting the structure of a microbial community [21] (Figure 1, Appendix A). Cells were clustered into gates with an average cell abundance per gate of 7658.1 cells. Following from this, due to low cell numbers per some of the sub-communities, three types of sub-communities were defined and named as dominant (DG), low abundant (LA), and outer cells (OG). The DG consisted of five sub-communities (G1, G2, G3, G4, and G9), while the LA sub-community consisted of 16 sub-communities (G5, G6, G7, G8, G10, G11, G12, G13, G14, G15, G16, G17, G18, G19, G20, and G21). Of the 200,000 cells per sample, DG accounted for 116,796, LA accounted for 44,024 and OG accounted for 39,180 cells. The cells in the DG sub-communities ranged between 8646 and 43,129, which was above the calculated average cell number per gate (7658.1) (Appendix A) (see Section 2.4). The cells inside the gates in LA ranged between 500 and 7311, below the defined threshold of 7658.1.

### 3.2. Quality of Recovered Metagenome-Assembled Genomes 

We recovered 24 MAGs of high and medium quality from all communities (sorted and unsorted). Three MAGs were retrieved from DG (two high-quality and one medium-quality), seven MAGs from LA (three high-quality and four medium-quality), and six from OG (two high-quality and four medium-quality). A total of eight MAGs were recovered from UC (six high-quality and two medium-quality) (Appendix A). The quality of MAGs recovered in sorted sub-communities differed in terms of the N50 statistics (ranging between 16–527), strain heterogeneity (varied between 0–100), and quality score (ranging between 51.41–93.04). The number of high-quality MAGs recovered from LA (3) was superior to the MAGs recovered from DG (two) and OG (two) (Appendix A). Furthermore, we mapped 72.38% reads from DG, 61.51% from OG, 58.92% from LA and 76.34% from the UC to the MAGs recovered in those libraries. This analysis indicates that other species exist in the community and were not recovered with our approach.

### 3.3. Taxonomic Profiling of MAGs

MAGs were classified as bacteria and placed in five different Orders. These MAGs were affiliated to 11 different taxa at different taxonomic levels based on average nucleotide identity (ANI) distances (Figure 2, Appendix A). Comparing the three sub-communities, the DG presented the least bacterial diversity with three different taxa: *Elizabethkingia ursingii* and *Escherichia flexneri*, classified up to species level, and *Sphingobacterium* sp., classified to genus level (Appendix A). The OG sub-community was composed of six taxa: *Sphingobacterium* sp., *Acinetobacter gerneri*, *E. flexneri*, *Comamonas terrigena* and *Empedobacter falsenii*, classified to species level, and *Variovorax* sp., classified to genus level. The LA sub-community presented the highest bacterial diversity consisting of seven taxa: *E.mpedobacter falsenii*, *E.scherichia flexneri*, *Elizabethkingia miricola*, *Acinetobacter pittii*, *C.omamonas terrigena*, *Sphingobacterium sp*., classified to species level, and *Variovorax* sp., classified to genus level (Appendix A).

Compared to the three sub-communities, the UC presented fewer taxa recovered. Six taxa were classified to species level (*Escherichia flexneri*, *C.omamonas terrigena*, *A. bouvetii*, *A. baumannii*, *Sphingobacterium* sp., *Empedobacter falsenii* and *Variovorax* sp.) (Appendix A). A total of five species were recovered in both the unsorted community and at least one or more of the sorted sub-communities. *E. flexneri* was present in all libraries (unsorted community and sorted sub-communities) (Appendix A). The number of gOTUs observed between the unsorted and sorted sub-community is shown in a Venn diagram (Figure 3).

### 3.4. Metagenome-Assembled Genomes (MAGs) Unique to Sorted Sub-Communities

Four MAGs (MDS_FC07—MDS_FC10) affiliated with four species were exclusively recovered in the sorted sub-communities (Appendix A). Of these, one MAG (classified as *Elizabethkingia ursingii*) from DG was of high quality with estimated completeness of 82.8%, contamination of 0.06%, and quality score of 82.50% (Table 1 and Appendix A). Two MAGs were exclusively retrieved in the LA sub-community: *Elizabethkingia miricola* and *A. pittii*. The completeness and contamination of *E. miricola* were 67.92% and 2.15%, respectively. *A. pittii* completeness was estimated at 51.41% complete and free of contamination (Table 1 and Appendix A). One medium-quality MAG (62.93% completeness and zero contamination), recovered exclusively in OG, was classified as *A. gerneri* (Table 1 and Appendix A).

### 3.5. Coverage and Relative Abundance of gOTUs per Library

We performed bin-relative abundance analysis using gOTUs recovered from the unsorted and sorted sub-communities to access the fraction of reads of gOTUs in each metagenome. After calculating the coverage, 11 taxa remained since one taxon (*C. B*-9) was found below the detection limit in all groups (Table 1) and removed from further analyses. The average detection limit of the gOTUs with coverage above ten times was 2.92% (±0.57). The rarefaction curves of genome coverage of the microbial communities observed in the unsorted and sorted sub-communities are illustrated in Figure 4, where all communities were able to achieve their plateau for the given cut-off coverage.

*Escherichia flexneri* was the only taxon recovered in all libraries. It was also the most dominant taxon detected in DG (55.47%) and OG (26.47%) but not in UC and LA. The most dominant species in UC and LA were *A. baumannii* (24.92%) and *Variovorax sp*. (30.90%). On the other hand, coverage of *Comamonas terrigena* was almost 2-fold lower in OG (5.67%) compared to LA (10.90%) and UC (10.63%). The gOTUs with the smallest coverage in the UC are *A. gerneri* and *Sphingobacterium* sp. (3.03%). Further, nine gOTUs were found below the detection limit in one or more sub-communities: six gOTUs in the DG, two gOTUs in the LA, and one gOTU in the OG. However, these taxa were found with coverage above the threshold in one or more sub-communities (Table 1) (see Section 3.1). The genome coverages and relative abundances obtained for gOTUs in each sub-community can be found in Table 1.

### 3.6. Reconstruction of the 16S rRNA Gene and Comparisons with gOTUs

We reconstructed 16S rRNA genes (rOTUs) from the unsorted and sorted sub-communities to compare their microbial community diversity to that of MAGs in all groups. A total of 57 rOTUs affiliated with 29 taxa at species level were generated from the entire dataset: 9 rOTUs from UC, 16 rOTUs from DG, 15 rOTUs from LA, and 17 rOTUs from OG. 

For rOTUs, a total of nine taxa were found in all groups: *Escherichia flexneri, A. bouvetii, A. baumannii, A. pittii, Acinetobacter bereziniae, Citrobacter pasteurii, Elizabethkingia miricola, Sphingobacterium multivorum,* and *Stenotrophomonas pavanii*. However, the relative abundances of these taxa varied in the different sub-communities (Appendix A). 

*Achromobacter insuavis*, *Bacillus tropicus,* and *Klebsiella granulomatis* were exclusively recovered in the OG. The relative abundances of these taxa were 0.59%, 0.51%, and 0.35%, respectively. One taxon (*Diaphorobacter ruginosibacter* with relative abundances of 1.05%) was retrieved only in the LA (Appendix A). *Brevundimonas olei* was uniquely found in the UC with a relative abundance of 1.66%. Moreover, three taxa were found only in the LA and OG. These taxa were *Acinetobacter gandensis* (relative abundance in LA 12.09% and relative abundance in OG 1.57%)***,***
*Delftia acidovorans* (relative abundance in LA 4.42% and relative abundances in OG 0.72% and 0.27%, respectively)**,** and *Enterobacter hormaechei* (relative abundances of 0.16% and 1.80% in LA and OG) (Appendix A).

Eight species (*Achromobacter anxifer, A. gerneri, Comamonas terrigena, Chryseobacterium artocarpi, Chryseobacterium geocarposphaerae, Klebsiella huaxiensis, Moraxella osloensis*, and *Pseudomonas qingdaonensis*) were recovered in multiple sorted and unsorted communities (Appendix A). The average detection limit of the relative abundance of recovered rOTUs was 0.15% (±0.20) (Appendix A). The complete relative abundances obtained for gOTUs and rOTUs in each sub-community can be found in Appendix A.

Eight of taxa found as gOTUs were also found as rOTUs (*Escherichia flexneri, Comamonas terrigena, A. bouvetii, A. baumannii, Empedobacter falsenii, Elizabethkingia miricola, A. pittii* and *A. gerneri*) (Appendix A). However, only *Escherichia flexneri* and *Comamonas terrigena* were found in the exact same sub-communities) in both gOTUs and rOTUs. Furthermore, the relative abundances of *Escherichia flexneri* and *Comamonas terrigena* were all greater than the average detection limit of relative abundance in all sub-communities for both gOTUs and rOTUs. *A. bouvetii* and *A. baumannii* rOTUs were present in all sub-communities, while they were only recovered in UC as gOTUs.

*Empedobacter falsenii* was recovered in all sub-communities as gOTU except DG. In contrast, *Empedobacter falsenii* rOTUs were only found in UC (Appendix A). Relative abundances of *Empedobacter falsenii* gOTUs and rOTUs were greater than the average detection limit in all sub-communities. *Elizabethkingia miricola* and *A. pittii* rOTUs were present in all sub-communities, with their relative abundances all greater than the detection limit (>0.15%). *Elizabethkingia miricola* and *A. pittii* gOTUs were only obtained in LA sub-communities. Although *Elizabethkingia miricola* gOTU was only recovered in LA communities, their relative abundances in OG and UC communities were greater than the average detection limit. In the case of *A. pittii*, the relative abundances in all sub-communities were all greater than the detection limit despite only being recovered in LA. *A. gerneri* was only found as gOTU in OG, while the rOTUs were found in all sub-communities except OG. Although *A. gerneri* gOTUs were only recovered in OG, their relative abundance in LA (2.81%) was greater than the detection limit. All *A. gerneri* rOTUs obtained in DG, LA and UC had a relative abundance greater than the average detection limit (1.87%, 0.61% and 5.33%, respectively). *Sphingobacterium* sp., *Elizabethkingia ursingii* and *Variovorax* sp.—classified gOTUs were not recovered in the rOTUs. *Sphingobacterium* sp. was not recovered in DG (Appendix A), but exhibited a relative abundance below the average detection limit of in all sub-communities. *E. ursingi* classified gOTUs were only recovered in DG, but their relative abundance was below the average detection limit (1.19%). In contrast, the relative abundance of *Elizabethkingia ursingii* gOTUs in all other sub-communities was greater than the average detection limit. *Variovorax sp*. classified gOTUs were recovered in all sub-communities except DG, with the relative abundances all above the average detection limit except for DG. 

A total of 21 taxa were only found in rOTUs, with *A. bereziniae*, *Chryseobacterium pasteurii*, *Sphingobacterium multivorum* and *Stenotrophomonas pavanii* being recovered in all subcommunities (Appendix A). The relative abundances of *Chryseobacterium pasteurii*, *Sphingobacterium multivorum* and *Stenotrophpmonas terrae* were greater than the average detection limit (>0.15%) in rOTUs, while *A. bereziniae* relative abundance was below the average detection limit in DG and LA sub-communities. Our data indicated that relative abundance, although relevant, may not be the most important feature for the recovery of MAGs.

## 4. Discussion

Our study combined high-throughput flow cytometry cell sorting and metagenomics to recover genomes in a cell culture of a wastewater microbial community. For this purpose, we performed Influx high-throughput cell sorting on the microbial community after separating cells virtually into sub-communities and sorting them via FACS before sequencing. By sorting cells of sub-communities, we could recover more gOTUs with higher quality than in whole communities. This phenomenon is related to the limited ability of metagenomics to recover low-abundant species [12,44]. By dividing the community into sub-communities, we reduced diversity in each sub-community but increased resolution in the entire sample. In this study, the 2-fold increase in the number of taxa identified in the sub-communities compared to the unsorted communities indicates the benefits of employing high-throughput cell sorting before metagenomics analysis. The number of taxa obtained in the sub-communities (11) further highlights the benefits of this strategy as, on average, only five MAGs are recovered per library [45], in contrast to the number that was obtained by our approach with 24 MAGS. 

In the present study, we recovered 16 MAGs from three sorted sub-communities (3 libraries), after staining the cells with DAPI and eight MAGs from the unsorted community (1 library). The study by Alteio et al. [20] also coupled metagenomics and flow cytometry cell sorting to improve genome recovery in soil communities. Alteio and collaborators sorted 359 ‘minimetagenome’ (sorted cells) after staining the cells with SYBR green, which can be difficult, and managed to recover 200 MAGs from the sorted cells. Furthermore, they retrieved 29 MAGs from the whole community (4 libraries). A possible reason for Alteio and collaborators to generate fewer MAGs per sorted cell group might be that the authors sorted the cells in microwell plates and applied metagenomes to individual sorted cells. While in our study, we combined the individual gates and generated only three sub-communities. Therefore, in the current study, we did not recover individual cells but created smaller sub-communities showing that these two studies had different approaches. We suggest that combining both approaches may recover a larger fraction of genomes in a microbial community.

The taxonomy of the genomes of different bacterial clades present in the wastewater treatment plant, including *Gammaproteobacteria* (six unsorted and nine sorted MAGs) and *Bacteroidia* (two unsorted and seven sorted MAGs), was classified at family, genus, and species levels. We found some species in these bacterial clades unique to sorted sub-communities. Interestingly, the number of taxa observed in the unsorted community and sorted sub-communities showed that some MAGs overlapped between all groups. For instance, *Escherichia flexnerii* was constantly present in all pooled sub-communities and the unsorted community, and three taxa (*Variovarax* sp., *Comamonas terrigena* and *Empedobacter falsenii*) found in the unsorted community were also recovered from the low abundant and outer sub-communities.

One species (*Escherichia flexneri*) was present in all communities and showed a high genome-coverage in all of the communities where it was recovered, indicating that *Escherichia flexneri* is one of the most abundant species in the community. Furthermore, we could recover several species (*Comamonas terrigena, Sphingobacterium* sp., *Empedobacter falsenii* and *Variovorax* sp.) across multiple communities; however, their coverages were, in some cases, below the detection limit. Interestingly, other species only found in a single community exhibited higher coverages in the libraries where they were not recovered (for instance, *A. bouvetti* and *A. baumanni*). We observed a similar trend for *Elizabethkingia ursingii*, which showed higher coverages in the communities where it was not recovered (in fact, it was below the detection limit in the DG sub-community where it was recovered). In other cases, species showed a higher coverage in libraries where they were not recovered as a MAG (e.g., *A. pitti* and *A. gerneri*). Some species showed coverages below the detection limit in the libraries where they were recovered (e.g., *Empedobacter falsenii* and *Elizabethkingia ursingii*). This trend suggests that coverage alone is not the primary driver in genome recovery. A study by Meziti and collaborators [46] used a minimum of 7x MAGs sequence coverage to quantify gene-level diversity within populations of microbial communities. In the current study, we were stricter and used 10x genome coverage. Another study by Meziti et al. [47] suggested 10x coverage is necessary for the reliable recovery of high-quality MAGs in the metagenomes. Our data strongly suggest that coverage is not the only factor influencing genome recovery because we could recover genomes with coverages below the detection limit in the libraries where species showed less than 10× genome coverage. For example, genomes affiliated with *Empedobacter falsenii* and *Elizabethkingia ursingii* were recovered as high-quality MAGs, with coverage of 5x in LA and 8x in DG.

A study by Liu and collaborators [21], using the same inoculum from our study and different cultivation conditions, recovered 23 amplicon sequencing variant (ASV) phylotypes classified to class level. The authors clustered their ASVs at 97% sequence similarity and considered ASVs with relative abundances below 0.35% contamination based on internal controls added to the sequencing run. A total of 22 classes of prokaryotes present in the study of Liu and collaborators were not found in our gOTUs. These classes were *Acidimicrobiia, Actinobacteria, Alphaproteobacteria, Anaerolineae, Bacilli, Blastocatellia, Betaproteobacteria, Caldilineae, Chlorobia, Chloroflexia, Clostridia, Cytophagia, Deltaproteobacteria, Flavobacteriia, Gracilibacteria, Halobacteria, Nitrospira, Thermotogae, Sphingobacteriia, ML635J-21, Verrucomicrobiae,* and *Saccharibacteria*. We reconstructed rOTUs from the metagenomic libraries of the unsorted and sorted sub-communities by binning the interleaved paired-end raw reads individually with the MATAM pipeline [37] to determine if there were other species within our data that we could not recover using gOTUs, because the reconstruction of rOTUs needs a smaller coverage in the metagenomes. We found 59 rOTUs classified to species level. In comparing species found in gOTUs and rOTUs, a total of eight taxa found in gOTUs were also found in the rOTUs. Still, not all shared the same distribution profile (i.e., not found in the same sub-communities). For example, *A. baumannii* gOTUs were only recovered in the unsorted sub-community (UC). In contrast, *A. baumannii* rOTUs were recovered in all sub-communities. By comparing our rOTUs to that of Liu and collaborators [21] we observed similar trends of microbial community compositions (at the class level) in the rOTUs found in the present study. Six classes of taxa found in our study were also present in the former study. However, 17 additional taxa classes found in the study by Liu and colleagues were not detected in our rOTUs. These classes were *Acidimicrobiia, Actinobacteria, Anaerolineae, Blastocatellia, Caldilineae, Chlorobia, Chloroflexia, Clostridia, Cytophagia, Deltaproteobacteria, Gracilibacteria, Halobacteria, Nitrospira, Thermotogae, ML635J-21, Verrucomicrobiae,* and *Saccharibacteria*.

The average detection limit of relative abundance of recovered rOTUs was 0.15% (±0.20) in the unsorted and sorted sub-communities, almost 20 times lower than the detection of our prokaryotic metagenome-assembled genomes (gOTUs; i.e., 2.92% ± 0.57). Five rOTUs (*Escherichia flexneri*, *A. baumannii*, *A. bouvetii*, *A. pittii* and *Elizabethkingia miricola*) were recovered in all sub-communities. Furthermore, these species have relative abundances above the rOTUs average detection limit in all groups (Appendix A). In the study of Liu and collaborators [21], the detection limit was 0.35%, two times higher than our rOTUs detection limit. It is important to note that Liu and collaborators had internal controls that were used to define the detection limit of the amplicon sequence variants in their studies. Studies involving internal controls for the recovery of metagenome-assembled genomes are lacking and would be necessary for determining the current limits of genome-centric analysis of microbial communities. 

Several factors may influence MAG recovery, such as sequencing depth, coverage, fragmentation, genetic diversity, and relative abundance of species [48,49]. Here, we studied the effect of species coverage and relative abundance in MAG recovery. Our results show that species coverage and relative abundance in MAG recovery do not fully explain our ability to recover MAGs and that other factors may play a more significant role. Moreover, other issues have been understudied in the recovery of MAGs, such as the presence of closely related species, which can lead to chimeras [50] and difficulties in species abundance estimation via the alignment of short reads. Sorting individual cells could be the better strategy to enhance the recovery of genomes of species with low abundances in a complex environment [12]. Furthermore, we observed that, on average, 32.71% (±8.39) of reads from each library were not mapped to any MAG. Thus, the extraction of higher yields and higher-quality DNA [51] may also help circumvent some of these issues. Integrating flow cytometry and metagenomics shows a higher resolution in deciphering microbial community composition by decreasing community complexity, including long-read sequencing data leading to improved genome-centric analyses of microbial communities.

## 5. Conclusions

Combining high-throughput cell sorting with metagenomics offers a promising opportunity to improve the genome recovery of low-abundant species from complex communities by reducing the complexity of environmental samples through cell sorting. Our study demonstrated that genome coverage and relative abundances are not the only essential factors governing the recovery of metagenome-assembled genomes, and further studies are necessary to underpin them (e.g., the presence of closely related species). Although we recovered species with genome coverage below the threshold (10x genome coverage), it would be important to test if those species were real with the addition of genome standards (internal controls) prior to sequencing. Furthermore, we suggest that combining single-cell sorting with separating sub-communities using flow cytometry may increase the recovery of underrepresented species in unsorted microbial communities.

## Figures and Tables

**Figure 1 microorganisms-11-00175-f001:**
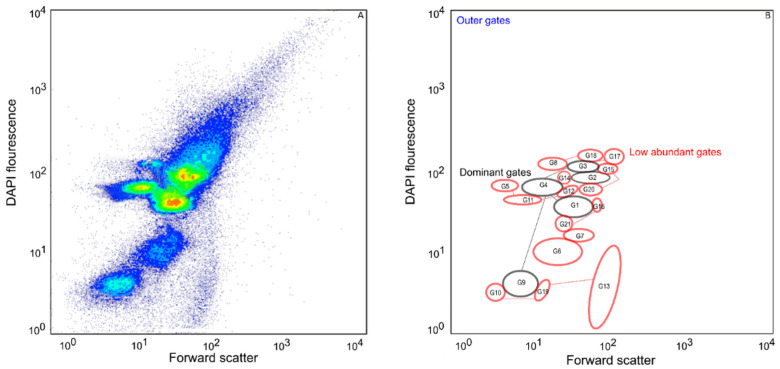
Visualization of microbial community dynamics. (**A**) Microbial community analysis based on fluorescence signals of cells. (**B**) Representation of gates for separated sub-communities. Dominant gates are delimited with black color (116,796 cells), low abundant gates are delimited with red color (56,776 cells) and outer cells are not in the delimited gate.

**Figure 2 microorganisms-11-00175-f002:**
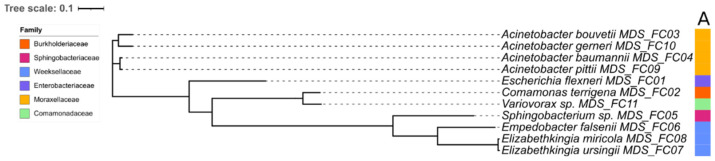
Phylogenetic tree of representative metagenome-assembled genomes using the concatenated protein alignment of genes from GTDB-tk [28]. The inside node labels represent taxonomy at the species level. The inner column (A), represents taxonomy at the family level. The size of the scale indicates relative evolutionary divergence value for taxa at each taxonomic rank.

**Figure 3 microorganisms-11-00175-f003:**
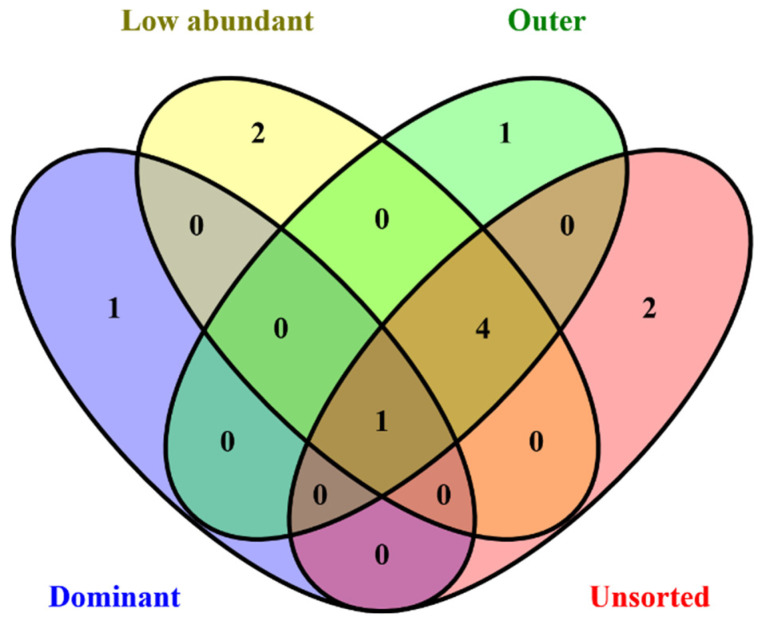
Venn diagram depicting the number of taxa observed shared between representative MAGs. It presents the number of taxa observed between MAGs recovered from unsorted and sorted sub-communities. One taxon was present in both the unsorted community and sorted sub-communities. Four taxa were found in the unsorted community and a combination of one or more sub-communities. Two taxa were present only in the unsorted community and five were exclusively found in the sorted sub-communities.

**Figure 4 microorganisms-11-00175-f004:**
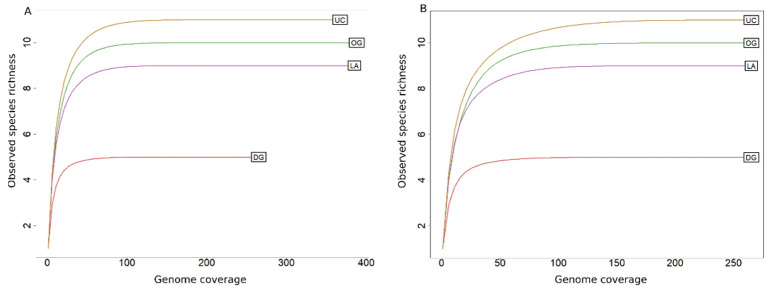
Rarefaction curves showing the observed time coverages of species in unsorted and sorted sub-communities. Reads were transformed to coverage before plotting the curves. (UC: Unsorted community; OG: Outer sub-community; LA: Low abundant sub-community; DG: Dominant sub-community; (**A**): before rarefaction; (**B**): after rarefying the data to 265 times genome coverage).

**Table 1 microorganisms-11-00175-t001:** Genome coverage of genome operational taxonomic units (gOTUs) per sub-community. All gOTUs in the same cluster were classified with the same taxonomy. Mapping the reads from all libraries to the gOTUs was done irrespective of their recovery. We removed gOTUs with less than ten times coverage and labeled them as below the detection limit (BDL). The marks indicate in which library the gOTUs were recovered. The numbers in parenthesis indicate the coverage times of each gOTU in the libraries expressed in percentages. (DG: Dominant sub-community; LA: Low abundant sub-community; OG: Outer gate sub-community; UC: Unsorted sub-community).

gOTUs	UC	DG	LA	OG	UC	DG	LA	OG	Species
gOTU_01	73 (22.8%)	147 (55.47%)	52 (13.50%)	103 (26.54%)	✓	✓	✓	✓	*Escherichia flexneri*
gOTU_02	35 (10.63%)	BDL (<3.80%)	42 (10.90%)	22 (5.67%)	✓	✗	✓	✓	*Comamonas terrigena*
gOTU_03	30 (9.11%)	45 (16.98%)	72 (18.70%)	89 (22.93%)	✓	✗	✗	✗	*Acinetobacter bouvetii*
gOTU_04	82 (24.92%)	28 (10.56%)	25 (10.56%)	41 (10.56%)	✓	✗	✗	✗	*Acinetobacter baumannii*
gOTU_05	10 (3.03%)	10 (3.77%)	BDL (<2.60%)	BDL (<2.58%)	✓	✗	✓	✓	*Sphingobacterium* sp.
gOTU_06	35 (10.63%)	BDL (<3.80%)	BDL (<2.60%)	13 (3.35%)	✓	✗	✓	✓	*Empedobacter falsenii*
gOTU_07	17 (5.16%)	BDL (<3.80%)	14 (3.63%)	16 (4.12%)	✗	✓	✗	✗	*Elizabethkingia ursingii*
gOTU_08	12 (3.64%)	BDL (<3.80%)	12 (3.11%)	11 (2.83%)	✗	✗	✓	✗	*Elizabethkingia miricola*
gOTU_09	43 (3.06%)	35 (13.20%)	35 (9.09%)	32 (8.24%)	✗	✗	✓	✗	*Acinetobacter pittii*
gOTU_10	10 (3.03%)	BDL (<3.80%)	14 (3.63%)	24 (6.18%)	✗	✗	✗	✓	*Acinetobacter gerneri*
gOTU_11	20 (6.07%)	BDL (<3.80%)	119 (30.90%)	37 (9.53%)	✓	✗	✓	✓	*Variovoarax* sp.

## Data Availability

The libraries used in this study are deposited in the National Center for Biotechnology Information (NCBI) with the accession numbers SRR17237133-SRR17237135. The gOTUs and rOTUs reconstructed in this study can be found in the following accession number JALBWX000000000-JALBXU000000000 and OP764601, ON081066-ON081112, OP764612, OP764656-OP764657, OP782786-OP782787, OP824856, OP837524, OP846957 and OP854891.

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
