# Peer review of "Combining Flow Cytometry and Metagenomics Improves Recovery of Metagenome-Assembled Genomes in a Cell Culture from Activated Sludge"

_microorganisms, 2023, doi:10.3390/microorganisms11010175_

Round 1
Reviewer 1 Report
This is a very interesting and important study, although the approach implemented by the authors is unlikely to be widely used in routine metagenomic researches due to its complexity. The article should certainly be published in the Microorganisms.
A small technical note: the full generic names of bacteria or their two-letter abbreviations should be used in the text, since the names of some genera begin with the same letters.
Author Response
We thank the reviewer for his comments. We have reviewed the entire manuscript and, for the exception of Acinetobacter bacteria, all others were replaced with their full names to avoid confusion.
Reviewer 2 Report
The paper is interesting, clearly written and the topic of the paper is very topical. Considering the use of a large number of abbreviations, it is necessary to include a list of the abbreviations used at the end of the paper to make it easier to follow. Given the number of analyses, it would be good to make a schematic diagram of the experiment and indicate the end point of the analysis. The genus and species of microorganisms are sometimes not written in italics, this should be corrected.
Author Response
We thank the reviewer for his comments. We have included a list of abbreviations at the end of the manuscript. We have also added a scheme showing the entire process and resulting analyses at each endpoint. This scheme was combined with Supplementary Figure 1. Supplementary Figure 1 is now comprised of Figure 1A and 1B, and its legend was updated to reflect the changes. The manuscript's main text was also updated in Lines 87-90 to include the mention of this scheme.